# Effects of interorganisational information technology networks on patient safety: a realist synthesis

Justin Keen ![ORCID],[1] Maysam Ali Abdulwahid,[1] Natalie King,[1] Judy M Wright ![ORCID],[1] Rebecca Randell ![ORCID],[2] Peter Gardner,[3] Justin Waring,[4] Roberta Longo,[1] Silviya Nikolova,[1] Claire Sloan,[1] Joanne Greenhalgh[5]

¹Leeds Institute of Health Sciences, University of Leeds, Leeds, UK
²Faculty of Health Studies, University of Bradford, Bradford, UK
³School of Pharmacy and Medical Sciences, University of Bradford, Bradford, UK
⁴Health Services Management Centre, University of Birmingham, Birmingham, UK
⁵Sociology and Social Policy, University of Leeds, Leeds, UK

**Correspondence to**
Dr Justin Keen;
J.Keen@leeds.ac.uk

## ABSTRACT

**Objective** Health services in many countries are investing in interorganisational networks, linking patients' records held in different organisations across a city or region. The aim of the systematic review was to establish how, why and in what circumstances these networks improve patient safety, fail to do so, or increase safety risks, for people living at home.

**Design** Realist synthesis, drawing on both quantitative and qualitative evidence, and including consultation with stakeholders in nominal groups and semistructured interviews.

**Eligibility criteria** The coordination of services for older people living at home, and medicine reconciliation for older patients returning home from hospital.

**Information sources** 17 sources including Medline, Embase, CINAHL, Cochrane Library, Web of Science, ACM Digital Library, and Applied Social Sciences Index and Abstracts.

**Outcomes** Changes in patients' clinical risks.

**Results** We did not find any detailed accounts of the sequences of events that policymakers and others believe will lead from the deployment of interoperable networks to improved patient safety. We were, though, able to identify a substantial number of theory fragments, and these were used to develop programme theories.

There is good evidence that there are problems with the coordination of services in general, and the reconciliation of medication lists in particular, and it indicates that most problems are social and organisational in nature. There is also good evidence that doctors and other professionals find interoperable networks difficult to use. There was limited high-quality evidence about safety-related outcomes associated with the deployment of interoperable networks.

**Conclusions** Empirical evidence does not currently justify claims about the beneficial effects of interoperable networks on patient safety. There appears to be a mismatch between technology-driven assumptions about the effects of networks and the sociotechnical nature of coordination problems.

**PROSPERO registration number** CRD42017073004.

## BACKGROUND

Many people who live in their own homes, and who have a number of health problems,

> **Strengths and limitations of this study**
>
> ► This is the first systematic review that seeks to explain the effects of extrahospital information technology networks on patient safety.
> ► The review investigates the distance between policy aspirations and realities in clinical settings.
> ► We were only able to test a limited set of possible explanations for the effects of networks.
> ► Aspects of quality appraisal rely on research team judgements: other teams might make different judgements.
> ► Breadth of coverage was maximised, and to some extent traded off against the overall quality of included articles.

need support from a range of professionals. There is good evidence that treatment and care is often fragmented, and increases patients' safety risks.[1–3] Policymakers and opinion leaders have argued that interoperable networks, which give clinicians access to patient records held in healthcare organisations across cities and regions, can help to overcome the fragmentation.[3 4] The networks should, therefore, support safer treatment and care.[5]

Health services have long had many discrete information technology (IT) systems, developed by different suppliers, so that general practitioners, community nurses, pharmacists and others use different systems. From the 1980s onwards hospital departments also had their own systems for pathology, radiology, operating theatres, and so on.[6–8] That is, the IT systems have also been fragmented, and in the views of policymakers also need to be integrated.[9 10] In practice, the technological task is to link the different systems together in an interoperable network that spans a geographical area, such as a city or county. The networks can be designed in different ways. At one end of a continuum, a network

provides clinicians with 'seamless' access to systems across a locality, so that they appear as a single patient record and are easy to navigate. At the other, clinicians can access the various different systems, with their own formats, and have to learn how to navigate each one to locate the information they need. It is not clear what types of network are currently available to clinicians. Systematic review evidence about the use of, and effects of, these networks is relatively limited and mixed.[11 12] Policy thinking therefore rests largely on assumptions about the value of interoperable networks.

This article reports on the findings of a systematic review, using the realist synthesis method, of the effects of interoperable networks on patient safety, which we defined as quantified changes in patients' clinical risks. The review included all configurations of networks, and effects could be attributable to networks or to a combination of networks and users. There were two strands to the review, focused on the coordination of services, and on the reconciliation of medication information, for older people living at home. The review identified policymakers and other stakeholders' assumptions about the ways in which interoperable networks influence clinical processes and outcomes and then evaluated the extent to which evidence about actual deployments supports the assumptions (or fails to do so).[13–16]

The realist synthesis included structured database searches of a wide range of databases, supplementary searches and stakeholder consultation. A range of quantitative and qualitative evidence was included. The synthesis is reported according to the Preferred Reporting Items for Systematic Reviews and Meta-Analyses (PRISMA) guidelines, and is consistent with the Realist and Meta-narrative Evidence Synthesis: Evolving Standards guidelines.[17 18] A protocol was developed and submitted to the PROSPERO registry of systematic reviews prior to commencing the review.

## DESIGN

The realist synthesis was undertaken in two discrete stages: theory development and empirical assessment. Theory development involved the development of programme theories, these being representations of the way(s) in which an intervention is intended to work.[19–21] Structured database and complementary searches were undertaken to identify published theories, or fragments of theories, that is, theories which covered a part of the sequences of decisions and actions that lead from the intervention to a safety-related outcome. Stakeholder consultation is usual in realist syntheses, and in this study nominal group consultation was used to refine the initial programme theories, and to help identify appropriate populations and settings for assessment. Assessment was undertaken through structured database and complementary searches, designed to identify empirical evidence to establish whether programme theories worked in the

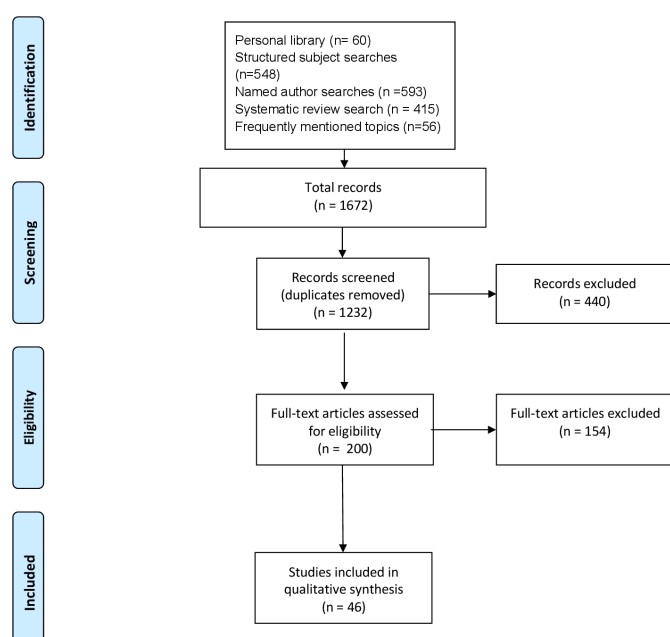

**Figure 1** Theory development Preferred Reporting Items for Systematic Reviews and Meta-Analyses (PRISMA) diagram.

ways that were intended—or did not work, or worked in some other way.

### Theory development

Five theory development searches were undertaken, including structured subject searches, a review of England and US government policies and official reports, named author searches (for Professors David Bates and Robert Wachter), systematic review and a Web of Science 'usage count' (articles with a high level of interest) searches (see figure 1). We were looking for statements that set out authors' reasoning about the effects of interoperable networks, which might be found in a wide range of texts, including editorials and interviews as well as journal articles and book chapters. They could be described using different terms, so we used very broad inclusion criteria and read substantial numbers of full texts. Passages where theories and fragments were described were copied into Word files, or hyperlinks created to long passages. The passages were synthesised into initial programme theories.[22]

We recruited two nominal groups of national policymakers and of IT leads responsible for interoperable networks in two regions of England. Consultation with senior managers at National Health Service (NHS) Digital led us to identify six national policymakers from three different organisations (NHS Digital, NHS England, Public Health England), to give us a range of perspectives. Five IT leads for different localities in the south of England were identified purposively as leaders in implementing interoperable networks by NHS colleagues in the north of England. We had no prior links with them. The initial theories, in the form of diagrams and supporting explanatory text, were presented to the nominal groups, and also to a patient and public involvement (PPI) panel.

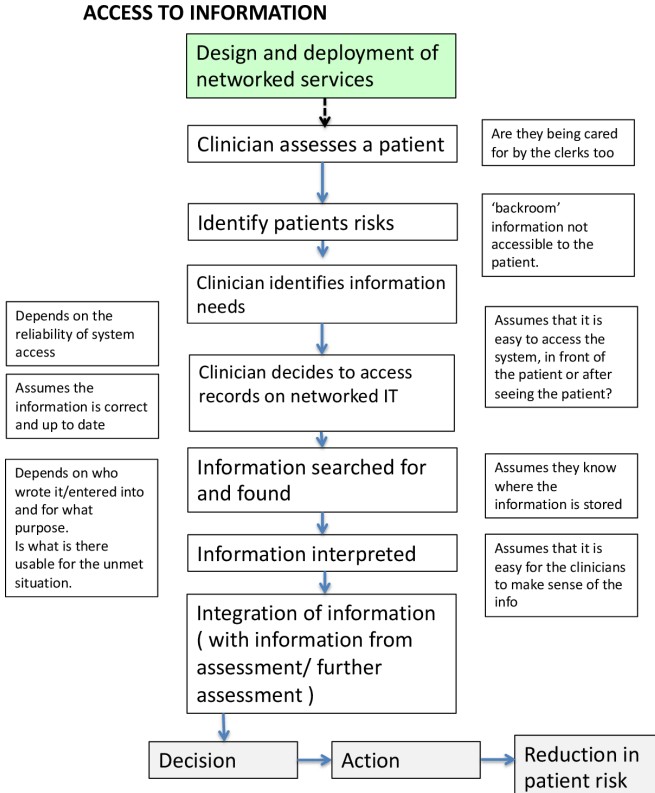

**ACCESS TO INFORMATION**

**Figure 2** Main programme theory. IT, information technology.

Their members commented on the face validity of the programme theories, suggested ways in which they might be refined and indicated their priorities for evidence searches. Their comments fed into the identification and development of a main programme theory, and to inform the design of detailed evidence searches.

The nominal groups indicated that, rather than fine-grained analyses of programme theories, they would value evidence about the effects on patient safety, and quality of treatment and care more generally. They observed there was little incentive to provide anything beyond functional—basic—interoperability, as many clinicians only used networks 'tactically', when they really needed information. As a result, evidence about functional interoperability would be valuable to them. They also indicated their priorities for evidence searches, leading us to settle on a single programme theory for assessment, and exploration of two functions of interoperable networks—in coordinating care across professional and organisational boundaries, and in medicine reconciliation (see figure 2). The PPI panel supported these priorities.

## Patient and public involvement

The PPI panel met three times in the course of the review and contributed in the following ways:

► Their comments helped us to set priorities for the populations and settings used in the searches.
► They contributed to the interpretation of the initial findings of the evidence searches.

► They commented on our overall interpretation of our findings at the end of the review.

This being a literature review, patients were not recruited to this study. Summaries and full reports of the review have been sent to the nominal and PPI groups.

## Evidence searches: data sources and searches

Searches were designed to interrogate the programme theory for two different functions of interoperable networks—the coordination of services for older people living in their own homes who were in receipt of two or more services from different organisations and, more specifically, the coordination of services for older people living in their own homes who had been prescribed medication from two or more organisations. (We focused on medicine reconciliation involving two or more lists of medications, and lists that were reconciled with a patient assessment undertaken by a clinician.) We undertook three searches for each function, focusing on the nature and extent of coordination problems (that interoperable networks might in principle help to address), on user experiences of using interoperable networks and on outcomes (defined as changes in patients' risks of harm).

The designs of the searches for each function were similar, allowing us to compare and contrast findings. All searches included the Medline database bundle 'Ovid MEDLINE and Epub Ahead of Print, In-Process & Other Non-Indexed Citations and Daily 1946-present'.[18] (Details of additional databases used in each search, and PRISMA flow diagrams are provided in separate files.) Search terms and synonyms were identified by the project team on the basis of a programme theory selected for assessment (see figure 2). All of the searches were performed and peer reviewed by information specialists (NK, JW). Structured search strategies were developing using free text words, synonyms and subject index terms organised into search concepts. Further complementary searches, including forward and backward citation searches, were also undertaken.

## Study selection

Inclusion and exclusion criteria were designed for each search, with the following inclusion criteria common to all searches:

► Written in the English language.
► Published in 2000 or later.[23]

Articles were assessed for relevance and rigour by three reviewers (MAA, JG, JK).

Relevance was assessed pragmatically using a 'target', akin to an archery target. Articles that met all inclusion criteria were placed in the 'bullseye'. Those that only partially met the population (eg, frail older people) criteria, but met other criteria, were placed in the middle ring. Articles that did not strictly meet either population or intervention criteria, but where it was judged that they might nevertheless shed light on the mechanisms involved, were placed in the outer ring.

## Data extraction and quality assessment

For included articles, study identifiers (author, publication year, country), information about the intervention and study methods (methods used, numbers and types of participants) and the evidence itself were recorded in customised spreadsheets. In relation to rigour, critical appraisal skills programme quality assessment checklists were completed by one member of the team (MAA) and reviewed by colleagues (JK or JG) to appraise systematic reviews, narrative and cohort studies.

## Synthesis

The empirical evidence for each function was used for two comparisons, namely the functions with one another, and each function with the programme theory.[24] The comparison with the programme theory allowed us to identify the assumptions which were and were not supported by the evidence.[25 26]

## RESULTS

Following study selection and assessment 46 studies were included. Many general statements were found which stated that interoperable networks would improve patient safety. For example, a 2016 report for the US Office of the National Coordinator for Health Information Technology stated that interoperable networks:

> can improve... safety by improving the timelines and completeness of important patient health information.[27]

Similarly, a 2016 report for the NHS in England recommended that the NHS should:

> ...ensure interoperability as a core characteristic of NHS Digital ecosystem—to support clinical care and to promote innovation and research.[28]

The policy argued that the objectives for interoperability included enabling integrated workflow, service redesign and clinical decision support. We did not find any detailed accounts that described or explained how they would produce safer diagnosis, treatment and care. A substantial number of theory fragments were, though, identified.[12 29–32] For example, it was argued that interoperable networks would make information available to clinicians wherever and whenever it was needed, enabling integrated workflow and clinical decision support.[28] There were also statements about challenges and risks, associated with a lack of common data standards, problems with interfaces and concerns about privacy.[22] These and other fragments were combined to create an initial programme theory.

### Evidence searches: care coordination

Five reviews were included on the nature of coordination problems in services for older people (table 1). The hand search of King's Fund reports produced two further reviews. The reviews were conducted in different academic traditions, and used different review methods, but produced broadly similar findings. There was good evidence that coordination problems were social and organisational in nature. Leadership, organisation cultures and trust influenced the effectiveness of coordination. Communication problems were also highlighted. These were typically characterised as a combination of task failures—such as failures to pass key information about a patient to a clinical colleague—and problems with conveying messages to colleagues with different professional backgrounds. ITs were rarely mentioned as having a role in either creating or solving problems.

We included four reviews and two primary studies from the evidence searches for user experiences of interoperable networks in the adult and older population (rather than older people specifically, table 2). A forward citation search from a 2013 review of the computer-supported cooperative work literature did not yield any further articles.[16] Most evidence was based on interviews, and therefore on subjective reports of behaviour rather than direct observational evidence of user experiences. Key details were missing from most reports, including information about interface characteristics and other features of the interoperable networks studied. This said, there was a consistent theme, indicating that interoperable networks were difficult to use, with problems reported in accessing networks, searching for and finding relevant information, and hence perceptions that the time costs of these activities were excessive.

Three articles were included from the searches for evidence about patient outcomes. All three were based on interviews with small samples of interviewees, reporting on experiences with networks with limited functionality. There was a common picture of difficulties encountered in accessing and using them, which led to broadly negative perceptions of their value.[33–35] Five review articles were identified as relevant in a subsequent broader search, for evidence about adult patients in general (table 3). They included quantitative evidence, of variable quality, about intermediate measures including adverse event and hospital readmission rates. The results were, broadly, positive in the sense that population rates were reported to reduce, implying the possibility of reductions in patients' risks. However, there were also some negative results, and none of the reviews included any quantitative evidence about the effects of interoperable networks on patient outcomes.

### Evidence searches: medicine reconciliation

For the nature of medicine reconciliation problems question, one systematic review, one other review and two observational studies were included (table 1). The overall quality of empirical evidence was reasonable. It indicated that the challenges were social and organisational in nature, the main one being that responsibility for reconciliation is not clear on the ground, particularly following discharge from hospital to home. As a result, responsibility fell between professionals, principally pharmacists,

**Table 1** The nature of coordination and of medicine reconciliation problems

| Author and year | Country | Method | Topic | Data type | Rigour | Relevance | Processes | Outcomes/errors |
|---|---|---|---|---|---|---|---|---|
| **Coordination** | | | | | | | | |
| Auschra (2018)[51] | – | Systematic review | Barriers to integrated care | Qualitative | 1 | 1 | Range of interpersonal and institutional issues | – |
| Threapleton et al (2017)[2] | – | Scoping review | Barriers and facilitators to coordination | Quantitative (E&O) and qualitative | 2 | 1 | Organisational and cultural features of coordination | – |
| Allen et al (2017)[52] | – | Narrative review | Transition from hospital to home | Qualitative | 1 | 1 | Negotiation and navigation of service user independence | – |
| Kirst et al (2017)[53] | – | Realist review | Conditions for effective team integration | Quantitative (E&O) and qualitative | 1 | 1 | Service use, patient and provider experience | Patient health status |
| Hudson et al (2014)[54] | – | Narrative review | Transition from hospital to home | Quantitative (O) and qualitative | 2 | 3 | Range of organisational and interprofessional issues | Readmission rates, user satisfaction with transition |
| Goodwin(2014)[55] | Seven countries | Synthesis of case study evidence | Models of integrated care | Qualitative | 2 | 1 | Range including flexibility of team working, effective communication, focus on user needs | Effective integration associated with improved user satisfaction |
| Goodwin (2013)[56] | England | Multisite case study | Evaluation of care coordination programmes | Qualitative | 2 | 1 | Range including organisational models, team cultures, engagement | – |
| **Medicine reconciliation** | | | | | | | | |
| Godfrey et al (2013)[57] | – | Scoping review | Medication management | Quantitative (E&O) and qualitative | 2 | 1 | Include time costs of, and responsibility for, reconciliation, communication problems | Polypharmacy, potentially inappropriate prescribing |
| Tommelein et al (2015)[58] | Europe | Systematic prevalence survey | Potentially inappropriate prescribing | Quantitative (O) | 2 | 3 | – | Polypharmacy, patient characteristics including advanced age |

Continued

**Table 1** Continued

| Author and year | Country | Method | Topic | Data type | Rigour | Relevance | Processes | Outcomes/errors |
|---|---|---|---|---|---|---|---|---|
| Hernandez (2017)[59] | USA | Interviews (community nurses) | Coordination of services | Qualitative | 1 | 1 | Interprofessional coordination, communication problems | Polypharmacy, medication errors, adverse events |
| Kennelty et al (2015)[36] | USA | Interviews (pharmacists) | Reconciliation posthospital discharge | Qualitative | 1 | 1 | Resources, communication, interprofessional relationships | – |

E&O, experimental and observational (evidence).

doctors and nurses. Reconciliation could also be viewed as an administrative task (rather than a safety-promoting one), and was not deemed to be important by some professionals. Communication problems emerged clearly as a theme.

Nine articles were identified that shed light on user experiences of medicine reconciliation.[36–44] These were a mix of scenario-based and field-based observational studies. The search for evidence about patient outcomes did not identify any quantitative evidence about changes in patients' risks. Results about proxy outcomes—changes in reconciliation error rates—were mixed. Some articles indicated that use of an interoperable network was associated with a measurable reduction in reconciliation errors. Others reported problems with using systems, resulting in no effects on error rates.

### Synthesis

We did not find any detailed published accounts that described the ways in which interoperable networks might improve patient safety (or increase patients' risks). For both functions, clinicians found that it was difficult to access networks, and to find and use relevant patient information. The evidence about outcomes, for both functions, was limited.

The programme theory assumed that information would be easy to find and use, and that patients' clinical risks would be reduced: the evidence did not support these elements of the theory. More generally, the programme theory was technology driven, assuming that the introduction of interoperable networks would lead to improved processes and outcomes.

### DISCUSSION

We did not find evidence that policymakers and opinion leaders have thought through the logic of their assumptions about interoperable networks. A key assumption, seemingly widely shared, is that clinicians need access to comprehensive data on their patients, wherever it is held. This has led to the policy prescription of interoperable networks in many countries. The prescription is intuitively reasonable, but our findings indicate that the underlying reasoning is flawed.

There were four main limitations to the study. First, we did not test alternative programme theories about the effects of interoperable networks on patient safety: it is not possible to rule out plausible explanations that have not been considered by policymakers and others, and hence by us. We might, for example, have drawn on complexity or other theories, instead of the views of policymakers and opinion leaders, to develop alternative programme theories. Second, realist synthesis is still developing as a systematic review method. The elements of our method have all been reported by other teams, but in slightly different combinations in different studies. Third, the aspects of the method involved team judgements. Other research teams might, for example,

**Table 2** Coordination of services: user experiences of interoperable networks

| Authors/year | Country | Methods | Topic | Rigour | Relevance |
|---|---|---|---|---|---|
| Eden et al (2016)[60] | – | Systematic review | HIE barriers and facilitators | 1 | 3 |
| Azarm-Daigle et al (2015)[61] | – | Systematic review | Cross-organisational data sharing | 1 | 2 |
| Hoerbst and Schweitzer (2015)[62] | – | Systematic review | Critical success factors for clinical information systems in integrated care | 1 | 2 |
| Wu and Larue (2015)[63] | USA | Systematic review | HIE barriers and facilitators | 1 | 3 |
| Nicolaisen and Berg (2015)[64] | Norway | Primary qualitative: interviews | Perceptions of messaging system | 1 | 2 |
| McMurray et al (2004)[65] | Canada | Primary qualitative: ethnographic study | Impact of partial interoperability | 1 | 1 |

HIE, Health Information Exchange.

adopt different criteria for assessing rigour and relevance. It seems reasonable to hope that their results would be broadly similar: our findings are, for example, consistent with reports of poor user experiences with IT systems in other settings.[45 46] But we cannot be sure of this. A fourth limitation is that we did not find studies of fully integrated networks, and so were not able to explore contributions of different network configurations to the observed process and outcome changes.

Technology-driven reasoning, based on the belief that the introduction of new networks will improve clinical processes and outcomes, appears to be widespread.[47 48] Our findings point to two problems with the reasoning. First, interoperable networks are usefully thought of as sociotechnical systems, where any effects result from combinations of technologies and users, rather than

the technologies alone.[49 50] Second, policy thinking implicitly assumes that networks will be easy to access and use. Our evidence indicates that this is not the case in practice, with many reports of difficulties. As far as we are able to tell, given sometimes limited information about interventions, all of the reported studies were conducted on functional networks. Clinicians were able to access other organisations' record systems, but had to navigate the different configurations in each one. We suggest that reports of difficulties accessing and using networks are not, therefore, surprising.

We conclude that policymakers and other stakeholders, including clinicians and suppliers, should examine the mechanistic assumptions that underpin current thinking. They should consider focusing on the everyday organisational realities of working across

**Table 3** Coordination of services: service and patient outcomes

| Authors/year | Country | Methods | Topic | Rigour | Relevance |
|---|---|---|---|---|---|
| King et al (2012)[33] | Scotland | Interviews | Electronic shared assessment tool | 1 | 3 |
| Waterson et al (2012)[34] | England | Interviews, observation of meetings | E-health supported care pathway | 2 | 3 |
| Vimarlund et al (2008)[35] | Sweden | Interviews | Virtual health record tool | 3 | 3 |
| Health Quality Ontario (2013)[66] | – | Systematic review | e-tools, HIE and care coordination | 1 | 2 |
| Sadoughi et al (2018)[67] | – | Systematic review | HIE, quality of care | 1 | 2 |
| Hersh et al (2015)[31] | – | Systematic review | Effectiveness of HIE | 1 | 2 |
| Reis et al (2017)[68] | – | Review of systematic reviews | Cost-benefit of records, HIE, interoperability | 1 | 2 |
| Menachemi et al (2018)[69] | – | Systematic review | HIE and service changes | 1 | 2 |

HIE, Health Information Exchange.

professional and organisational boundaries, and ways in which networks might make it easier for busy clinicians to coordinate their work with one another. Looking ahead, these findings highlight a puzzle about the quality of interoperable networks available to clinicians: there is a need to understand why this continues to be the case. To this end, there is merit in developing policies which set out sociotechnical cases for investments in detail. In addition we, in common with other authors, believe that primary field studies of user experiences of interoperable networks, to identify solutions more acceptable to clinicians, are needed.[16]

**Acknowledgements** We are grateful to the members of the nominal groups, and the PPI panel, for their contributions to the design and conduct of the review.

**Contributors** PG, JG, JK, RL, RR, JW and JMW developed the proposal for the study. NK and JMW designed and undertook structured database searches. MAA, JG and JK undertook screening and data extraction. PG and JW provided specialist input to the design and interpretation of specific searches. All authors, including SN and CS, contributed to the detailed study design and to the interpretation of overall findings. All authors either drafted or commented on drafts of this article.

**Funding** National Institute for Health Research-Health Services and Delivery Research programme (project 16/53/03).

**Competing interests** None declared.

**Patient consent for publication** Not required.

**Ethics approval** University of Leeds Faculty of Medicine and Health Ethics Committee (MREC 17-004).

**Provenance and peer review** Not commissioned; externally peer reviewed.

**Data availability statement** The majority of material in this article is derived from already published articles and reports. Beyond this, other methods were qualitative and the data generated are not suitable for sharing. Further information can be obtained from the corresponding author.

**ORCID iDs**
Justin Keen http://orcid.org/0000-0003-2753-8276
Judy M Wright http://orcid.org/0000-0002-5239-0173
Rebecca Randell http://orcid.org/0000-0002-5856-4912

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
