## [Reviewer comments · BMJ Open]

ARTICLE DETAILS

TITLE (PROVISIONAL)	The effects of inter-organisational information technology networks on patient safety: a realist synthesis
AUTHORS	Keen, Justin; Abdulwahid, Maysam; King, Natalie; Wright, Judy; Randell, Rebecca; Gardner, Peter; Waring, Justin; Longo, Roberta; Nikolova, Silviya; Sloan, Claire; Greenhalgh, Joanne

VERSION 1 – REVIEW

REVIEWER	Amina Tariq QUT, Australia
REVIEW RETURNED	09-Feb-2020

GENERAL COMMENTS	Background Line 13: Reference not consistent in formatLine 22: The sentence need references to support the presented argument. Also the second sentence of the same paragraph is not clear.It would be valuable to add some information in the background on examples of studies that highlight evidence for inter-operability across healthcare organizations globally to substantiate the rationale of the systematic review.Line 55-56 : The scope of systematic review is unclear, this sentence presented as a quantitative systematic review where as the abstract highlights the qualitative nature. Couple of clear sentences on scope of the systematic review would be helpful.Line 54-56: It would be helpful for the readers if authors can clarify what do they mean by “effects” within the scope of their review.It is not clear if by interoperable networks authors mean interoperable technological networks our systems only. Design: How were the members of two nominal groups selected?How was the CASP assessment carried out? Results: Authors suggest that they were many general statements about interoperable networks - a few Examples of such statements would be valuable to understand this argument.It may be valuable to add the second paragraph of the results section in the methods/design as part of the synthesis process described. Discussion: It would be valuable to extend the discussion to examine implications of this review on how both policymakers and researchers understand interoperable networks. Perhaps
--

	highlighting the different levels of interoperability that sometime are beyond technology only are presumed, When describing their benefits. 2. It is difficult as a reader to understand the scope with which interoperable networks were examined within the systematic review, and that is reflected in the discussion as well where it seems that interoperable networks are considered only in terms of integrated electronic systems. It is therefore important to describe a definition of interoperable networks considered in scope of this review.
--	--

REVIEWER	Andrew Georgiou Macquarie University, Australia
REVIEW RETURNED	06-Apr-2020

GENERAL COMMENTS	This is a very interesting and thorough piece of work. It provides some valuable insights and raises important challenges about existing technological assumptions. I have the following comments and suggestions. Design (Page 10) – I would like to know more about the stakeholder consultation. Can the authors insert some details about who was involved in the consultation and how many stakeholders were involved? It is important to know what perspectives they brought to the process. Synthesis (Page 12) – The synthesis part of the study is important and is an obvious strength of the realist approach. However, apart from the paragraph on page 12, it is not clear what role the synthesis played in shaping the Results. I would like to see some more connections made through the Results and Discussion. Discussion (Page 21) – There is a strong evidence-based argument made against the assertion that improvements in technologies will be followed by improvements in treatments and outcomes. But I am puzzled by what exactly the authors mean by “IT solutions”. There are a lot of “IT solutions” out there. I am not even clear that the authors have provided a strong enough perspective on what they see as an “interoperable network” – particularly given that they were able to point to major inconsistencies in the evidence. Surely, the notion of interoperability has changed over time. It doesn’t necessarily mean the same as it did a decade ago. Does the evidence indicate this? My other concern about the argument mounted in the Discussion is that it describes the coordination of services as divided between socially-driven factors and technologically driven factors. Socio-technical approaches (which have a strong theoretical foundation in health informatics literature) seemingly did not make it into the realist synthesis. Not sure why not? So when the authors suggest that “clinicians need to communicate with and understand one another ...” a socio-technical approach could well contend that communication (be it by phone, fax, SMS, paper etc., or “interoperable networks”) generally involves some form of technology. Finally, I was also surprised that “complexity” theory did not get some more attention, particularly in regard to the limited amount of evidence about safety-related outcomes associated with interoperable networks. The problem of identifying and measuring concrete outcomes has been a perennial problem in computing generally and with digital health more specifically. Complexity theory could contend that a cause-effect approach to monitoring
---

	outcomes can, in some cases, be over-simplistic. Instead, it may be more accurate to suggest (which the authors allude to) that there are a complex array of factors (organisational, socio and technical and system-wide) that need to be considered.
--	--

VERSION 1 – AUTHOR RESPONSE

REVIEWER 1

Background – paragraph 2 (lines 13 and 22)

Agree. Paragraphs 2 and 3 of Background re-written, to address this and later comments.

Background – evidence to substantiate rationale for systematic review

Agree. Background re-drafted to make it clear that policy makers and opinion leaders are making *claims* about interoperable networks. We sought to identify and evaluate the claims. In the event, we did not find evidence that policy makers have systematically worked out a rationale for their claims.

Background – specify scope of review

Agree, minor edit in Abstract, text edited in new paragraphs 2 and 3 of Background, and edits in Methods.

Background – clarify ‘effects’

Agree, edited paragraphs 2 and 3 of Background.

Background – technological vs. socio-technical networks

Agree. This point overlaps with a similar one made by Reviewer 2 (see below). Our view is that the most widely used definitions – from the IEEE and HIMSS – confound technology-only and socio-technical perspectives. We suggest that there isn’t room to go into this wider issue in the article, but we have edited the Background text that seeks to make our focus clear.

Design – nominal group recruitment

Agree - text added

Design – CASP assessment

Agree - text added

Results – add examples of general statements

Agree – statements added

Results – move paragraph to Design section

Agree – moved to Design section

Discussion – policy makers' understanding of networks

Agree – as above, paragraphs 2 and 3 in Background edited, and Discussion also edited.

Discussion

Agree – as above, text in Background edited to clarify scope of review.

REVIEWER 2

Design – more information about stakeholder consultation

Agree, text added.

Results/Discussion – explain role of synthesis

Agree, text was missing, now added in Results.

Discussion – “IT solutions”

Agree, poor drafting in the relevant passages. “IT solutions” is not a helpful term. Term removed and text edited.

Discussion – technological vs socio-technical

Agree – more poor drafting. Text edited substantially in this part of the Discussion to make the socio-technical point more clearly.

Complexity theory

This is an interesting observation. Keen was a lead investigator in a Research Council programme (Large-Scale Complex IT Systems, EPSRC F001096/1), and has recently returned to the topic, to write an article on the relationship between systems and complexity theories. Greenhalgh and Wright have considerable experience of Realist methods. Based on our collective experience, our view is that Realist methods do not address issues raised by current thinking on complexity, notwithstanding Ray Pawson's interesting attempt to address it in his 2013 book, *The Science of Evaluation*. We suggest that this is too broad an issue to deal with properly in this article, but we have added a 'methods limitation' that acknowledges this comment.

VERSION 2 – REVIEW

REVIEWER	Andrew Georgiou Australian Institute of Health Innovation
REVIEW RETURNED	07-Jul-2020
GENERAL COMMENTS	Thank you for spending the time to carefully address reviewer comments.